# Training and Evaluating Norwegian Sentence Embedding Models

**Bernt Ivar Utstøl Nødland**

Norwegian Defence Research Establishment

Instituttveien 20

2007 Kjeller

`bernt-ivar-utstol.nodland@ffi.no`

## Abstract

We train and evaluate Norwegian sentence embedding models using the contrastive learning methodology SimCSE. We start from pre-trained Norwegian encoder models and train both unsupervised and supervised models. The models are evaluated on a machine-translated version of semantic textual similarity datasets, as well as binary classification tasks. We show that we can train good Norwegian sentence embedding models, that clearly outperform the pre-trained encoder models, as well as the multilingual mBERT, on the task of sentence similarity.

## 1 Introduction

Recently there have been a huge increase in the capabilities of natural language processing systems. The new dominant paradigm is using large language models such as BERT (Devlin et al., 2019) or GPT (Radford et al., 2018) as a starting model which one adapts to any given task one wishes to solve. There exists several different versions of BERT-type encoder models in Norwegian (Kummervold et al., 2021), (Kutuzov et al., 2021), (Pyysalo et al., 2021). It is well-known that BERT-type models that give contextual words embeddings do not give particularly good sentence embeddings (Reimers and Gurevych, 2019). For this reason we train and evaluate Norwegian sentence embedding models, using the pre-trained encoder models as starting points.

We train models using the state of the art SimCSE methodology, similarly to the original paper (Gao et al., 2021). Like them, we train both unsupervised and supervised models. We start with a pretrained bidirectional language encoder model such as BERT or RoBERTa (Liu et al., 2019). For the unsupervised version we sample texts from the Norwegian Colossal Corpus (NCC) dataset (Kummervold et al., 2022). We then pass them through the model using two different dropout masks and predict contrastively which pairs within a batch represent the same text. For the supervised version, we train on a machine-translated version of natural language inference (NLI) data, where we use sentences related by "entailment" as positive sentences, and sentences labeled as contradiction as hard negative sentences. We train on both the Norwegian dataset, and a combined dataset of both Norwegian and English NLI data, and show that the latter gives better results for sentence representations in Norwegian. We evaluate our models on a machine translated version of semantic textual similarities (STS) datasets, as well as on the sequence classification problems in Norwegian "Talk of Norway" and the binary classification version of the NoReC review dataset (Velldal et al., 2018).

Our main contributions are:

1. We train and evaluate Norwegian unsupervised and supervised sentence embedding models.

2. We demonstrate a new way to compare the various existing Norwegian language models by measuring their performance after training them to make sentence embeddings.

3. We show that our sentence encoders sometimes get better performance than the base encoder on classification . In particular, we obtain new state of the art results on the classification problem "Talk of Norway".

4. Through our experiments we illustrate the usefulness of machine translated datasets for training and evaluating Norwegian language models. In particular, we show that supervised training on machine translated data out-

performs unsupervised training on Norwegian data.

## 2 Related work

The fundamental technique we build on is that of training large transformer models (Vaswani et al., 2017). In particular, we utilize the large encoder models Bidirectional Encoder Representations from Transformers (BERT) and Robustly Optimized BERT (RoBERTa) by using them as pre-trained starting points.

Our work builds upon existing language models trained in Norwegian. The National Library of Norway has trained BERT models in Norwegian (Kummervold et al., 2021), which we call NB-BERT, which exists in both base and large size. Also, the language technology group at the University of Oslo has trained their version of a BERT for Norwegian called NorBERT (Kutuzov et al., 2021). There is also a WikiBERT model trained on Norwegian Wikipedia (Pyysalo et al., 2021). We also test the multilingual version of BERT (Devlin et al., 2019), which is trained in Norwegian and many other languages.

Our work uses existing methodology for making sentence embedding models. The first paper to improve BERT to make better sentence representations by training it for that purpose, was the Sentence-BERT paper (Reimers and Gurevych, 2019), which trained sentence embedding models by using siamese networks. We build upon the newer Simple Contrastive learning of Sentence Embeddings (SimCSE) methodology (Gao et al., 2021), which uses a contrastive training objective to create sentence embeddings from a pre-trained encoder. The idea behind both of these works is that of finding a training procedure that better extracts the knowledge about sentences that already exists in the pre-trained encoder model.

Most existing work in the literature on making sentence embeddings are either in English or uses multilingual models. Examples of the latter are mBERT and several other approaches such as (Feng et al., 2022), (Goswami et al., 2021) and (Reimers and Gurevych, 2020).

## 3 Data

For the unsupervised models, we sample data from the Norwegian Colossal Corpus (NCC) (Kummervold et al., 2022). This is a dataset of different smaller Norwegian text corpuses that has been col-

---

**Sentence:** Deltakerne mente at hvis interessenter var seriøse om å forbedre finansrapporteringsmodellen, ville en gruppe bli opprettet og finansiert spesielt for dette formålet. [Translation: Participants believed that if stakeholders were serious about improving the financial reporting model, a group would be created and funded specifically for this purpose.]
**Positive:** Deltakerne forventer at seriøse interessenter vil danne en gruppe for å forbedre finansrapporteringsmodellen. [Translation: The participants expect that serious stakeholders will form a group to improve the financial reporting model.]
**Negative:** A group was created to improve the financial reporting model.

Figure 1: An example of a triplet of sentences of mixed language in the Norwegian/English NLI dataset.

lected into one corpus by the National Library of Norway to train language models. This is primarily a Norwegian corpus, although there are some amounts of other languages present. The dataset description estimates that 87% of documents are in Norwegian, with about 6-7 % of documents in English and the rest in other European languages (mostly other Nordic languages). We sample 1 million texts from the dataset for training unsupervised. Some are longer than one sentence, but all are truncated to max 32 tokens before training, thus they are all approximately sentence length.

For supervised training we train with data collected for the task of natural language inference (NLI). This task is that of taking a pair of sentences and predicting the relationship between them as either "entailment", "neutral" or "contradiction". The authors of the SimCSE paper use NLI data to create triples of a sentence with one positive and one hard negative and show that this data work well for training sentence models using contrastive learning, thus we follow this practice. We use a dataset that has been curated for training in Norwegian by the National Library of Norway.[1] The original data is based on the English datasets the Stanford Natural Language In-

---

[1] https://huggingface.co/datasets/NbAiLab/mnli-norwegian

> **Sentence 1:** en mann skjærer opp en agurk .
> [Translation: a man cuts open a cucumber .]
> **Sentence 2:** en mann skjærer en agurk .
> [Translation: a man cuts a cucumber .]
> **Similarity:** 4.2
>
> **Sentence 1:** en mann spiller harpe .
> [Translation: a man plays the harp .]
> **Sentence 2:** en mann spiller et keyboard .
> [Translation: a man plays a keyboard .]
> **Similarity:** 1.5

Figure 2: Examples from the translated STS-Benchmark dataset. Similarity ratings are from 0-5.

ference (SNLI) Corpus (Bowman et al., 2015) and Multi-Genre Natural Language Inference (MNLI) dataset (Williams et al., 2018). The Norwegian data is machine translated from the MNLI dataset and has about 128 thousand triples. There is also a combined Norwegian and English version of the dataset made by taking a combination of the translated Norwegian MNLI data and English MNLI and SNLI data.[2] Also included are extra combined Norwegian/English sentence triples: For each of the translated triples there is a joint Norwegian/English triple consisting of one or two sentences in each of English and Norwegian, see Figure 1 for an example. The English/Norwegian dataset contains about 531 thousand triples of sentences.

For evaluation we also machine translate the standard English datasets for semantic textual similarity STS12-16 (Agirre et al., 2012), (Agirre et al., 2013), (Agirre et al., 2014), (Agirre et al., 2015), (Agirre et al., 2016), STSBenchmark (Cer et al., 2017), and SICK relatedness (Marelli et al., 2014). The task is predicting how similar a pair of sentences are to each other on a scale of 0-5. We use these datasets only for validation and testing and never for training. In fig. 2 we see two examples from the translated STS Benchmark dataset.

The usage of translated datasets is a weakness compared to having original data in Norwegian. This project can also be viewed as an exploration of what performance it is possible to get

from auto-translated English datasets: To the degree they are shown to be useful, one will have much more data one could potentially work with in Norwegian language processing. We note that for sentence similiarity, a similar exploration of translated data has been done for Swedish in (Isbister and Sahlgren, 2020). They conclude that they do not recommend the usage of automatically translated STS datasets for fine-tuning, but that it should probably have limited negative consequences for comparing models. We partly follow their recommendation: We only use translated STS data for valdiation and evaluation, but we do perform supervised training on translated NLI data.

## 4 Experiments

Our experiments follow the implementations in the SimCSE paper closely. We start with a pre-trained encoder model that is either BERT or RoBERTa.

For unsupervised training we sample one million texts from the NCC dataset. We then pass each text through the model using two different dropout masks to obtain two different text representations $s_i$ and $s_i^+$ for each text. Here dropout functions as a form of continuous augmentation of embeddings. Then we contrastively predict which pairs of texts within a batch are the same using cross-entropy loss on the cosine similarity scores. In other words, the loss for text $i$ is given by

$$\text{loss}_i = -\log \frac{e^{\text{sim}(s_i, s_i^+)/\tau}}{\sum_{j=1}^{b} e^{\text{sim}(s_i, s_j^+)/\tau}},$$

where sim is cosine similarity and $\tau$ is a temperature hyperparameter which we simply set to 0.05, which is the outcome of optimization done in the SimCSE paper.

For training unsupervised models, the models we start from are given by their names on huggingface as

- bert-base-cased [english model]

- roberta-base [english model]

- bert-base-multilingual-cased

- TurkuNLP/wikibert-base-no-cased

- ltgoslo/norbert2

- NbAiLab/nb-bert-base

---

[2] The same English data that was used to train English SimCSE: https://huggingface.co/datasets/princeton-nlp/datasets-for-simcse

| Model | Avg. STS |
|---|---|
| BERT | 34.29 |
| RoBERTa | 25.56 |
| mBERT | 48.34 |
| WikiBERT | 42.21 |
| NorBERT | 54.42 |
| NB-BERT-base | 50.41 |
| NB-BERT-large | 49.90 |

Table 1: Average performance of models before training using average of the last layer on Norwegian STS.

- NbAiLab/nb-bert-large

The english models are included as a sanity check: Since we are using automatically translated datasets to choose the best models, we want to compare their performance with some models that are expected to perform worse than Norwegian models. For the same reason we also test on the English STS datasets.

We train the supervised models using NLI data where each sentence has one paired sentenced labeled as entailment, which is regarded as a positive sample, and one sentence labeled with contradiction, which is considered a negative sample. We thus obtain three different sentence representations $s_i, s_i^+, s_i^-$. As in the SimCSE paper, we train contrastively trying to predict the positive pairs, and add the negative sentence representation $s_i^-$ to the loss function as follows:

$$\text{loss}_i = -\log \frac{e^{\text{sim}(s_i, s_i^+)/\tau}}{\sum_{j=1}^b e^{\text{sim}(s_i, s_j^+)/\tau} + e^{\text{sim}(s_i, s_j^-)/\tau}}$$
(1)

For training supervised models we start with the following models:

- bert-base-multilingual-cased

- TurkuNLP/wikibert-base-no-cased

- ltgoslo/norbert2

- NbAiLab/nb-bert-base

- NbAiLab/nb-bert-large

We train with the same settings as in the SimCSE paper: We set a max sequence length of 32, and use the learning rates and batch sizes given in the appendix of the SimCSE paper (which vary by model type and size). Each model is trained on a single NVIDIA 3090 GPU. For some models we have to use gradient accumulation to achieve the correct batch size due to lack of RAM, which changes training dynamics a bit, since contrastive loss depends on the entire batch. We do not see any noticable effects on results from this. We train with the Adam optimizer with linear weight decay and put a multi-layer perceptron (MLP) on top of the model for training. Unsupervised we train for one epoch, and supervised for three. The best model is selected by evaluating on the dev part of the STS Benchmark dataset. For evaluation we test both with and without this MLP, and find that generally, testing without the MLP gives slightly better results. We train three versions of each model and report average scores.

The models are also fine-tuned on two Norwegian sequence classification tasks. Talk of Norway (ToN) is a subset of the Norwegian parliament speeches dataset (Lapponi et al., 2018), where the task is to classify whether the speech was given by SV or FrP (politically left or right, respectively) selected in (Kummervold et al., 2021).[3] NoReC is a dataset of reviews in Norwegian from different domains such as movies, video games and music (Velldal et al., 2018). From this dataset one can extract a binary classification task by taking the subset of reviews that are clearly positive or negative and letting the task be to classify them as positive or negative (Øvrelid et al., 2020). We take the text representations made by the model before the MLP, and add a linear classification layer on top and fine-tune the entire model on the training dataset. For both the fine-tuning datasets we do a grid search for hyperparameters under the following conditions (these are the same hyperparameters as in the finetuning examples in the appendix of the original BERT paper (Devlin et al., 2019)):

- epochs=2, 3, 4

- learning rate = 2e-5, 3e-5, 5e-5

- batch size 16, 32

We use the macro f1 score on the validation set to select the best model for each training run. We do three training runs and report the average of test scores.

---

[3]https://huggingface.co/datasets/NbAiLab/norwegian_parliament

| Model | STS12 | STS13 | STS14 | STS15 | STS16 | STSB | SICKR | Avg. |
|-------|-------|-------|-------|-------|-------|------|-------|------|
| BERT | 55.21 | 49.64 | 49.29 | 63.68 | 54.39 | 54.67 | 50.93 | 53.97 |
| RoBERTa | 60.30 | 59.12 | 57.15 | 68.73 | 64.33 | 64.04 | 54.39 | 61.15 |
| mBERT | 60.88 | 62.31 | 55.91 | 70.78 | 66.80 | 61.87 | 57.13 | 62.24 |
| WikiBERT | 63.38 | 70.21 | 62.63 | 74.04 | 70.90 | 70.88 | 62.52 | 67.79 |
| NorBERT | 56.41 | 65.33 | 54.32 | 68.95 | 68.00 | 62.40 | 64.54 | 62.85 |
| NB-BERT-base | 59.40 | 70.70 | 57.93 | 71.87 | 69.94 | 69.25 | 63.98 | 66.15 |
| NB-BERT-large | **70.45** | **80.80** | **72.79** | **81.53** | **78.41** | **79.35** | **69.18** | **76.07** |

(a) Performance of unsupervised models on the Norwegian STS datasets.

| Model | STS12 | STS13 | STS14 | STS15 | STS16 | STSB | SICKR | Avg. |
|-------|-------|-------|-------|-------|-------|------|-------|------|
| mBERT | 73.43 | 69.09 | 70.84 | 81.50 | 73.82 | 76.47 | 72.79 | 73.99 |
| WikiBERT | 73.29 | 64.48 | 69.24 | 80.32 | 74.51 | 75.42 | 69.94 | 72.45 |
| NorBERT | 74.30 | 70.69 | 72.09 | 82.56 | 76.91 | 79.33 | 73.74 | 75.66 |
| NB-BERT-base | 76.31 | 77.20 | 75.43 | 84.47 | 77.69 | 82.14 | 77.97 | 78.75 |
| NB-BERT-large | **77.07** | **83.65** | **80.28** | **86.24** | **81.87** | **84.37** | **78.44** | **81.70** |

(b) Performance on the Norwegian STS datasets of supervised models trained on both Norwegian and English NLI data.

| Model | STS12 | STS13 | STS14 | STS15 | STS16 | STSB | SICKR | Avg. |
|-------|-------|-------|-------|-------|-------|------|-------|------|
| mBERT | 69.28 | 71.50 | 69.44 | 78.12 | 74.38 | 71.12 | 67.70 | 71.65 |
| WikiBERT | 70.14 | 71.18 | 71.79 | 77.56 | 76.20 | 74.20 | 67.32 | 72.63 |
| NorBERT | 70.79 | 74.46 | 72.44 | 80.66 | 77.73 | 76.65 | 71.56 | 74.90 |
| NB-BERT-base | 72.41 | 79.22 | 74.67 | 81.47 | 77.72 | 78.49 | 73.50 | 76.78 |
| NB-BERT-large | **74.67** | **83.65** | **79.47** | **84.15** | **81.82** | **82.25** | **74.75** | **80.11** |

(c) Performance on the Norwegian STS datasets of supervised models trained on Norwegian NLI data.

Table 2: Results of our models tested on the Norwegian STS datasets(Spearman's correlation).

## 5 Results sentence similarity

We evaluate the trained models on the semantic textual similarity datasets. We evaluate our models both on the Norwegian version of the datasets, and the original English. We report Spearman's correlation for the STS datasets.

### 5.1 Evaluation in Norwegian

In Table 1 we see the average performance on the Norwegian STS before training using the average of the last layer to compare embeddings. We also tested using the average of first and last layers (giving similar numbers) and using "cls" token (giving worse numbers). Thus we have a baseline to compare how much the models have learned from the training.

In Table 2a we see the performance of our unsupervised models on the Norwegian STS datasets. These are the results when we test without the MLP, which on average performs slightly better than using MLP also for testing.

In Table 2b we see the results from training supervised models on the combination of Norwegian and English NLI data, while Table 2c shows

the performance when training on only Norwegian NLI data. We see that training with English included improves performance over merely training in Norwegian for all models.

We see that the supervised models perform much better than the unsupervised ones. This would usually not be surprising, but considering the supervised data is automatically translated and therefore presumably of lower quality than the unsupervised data, it is interesting to note.

### 5.2 Evaluation in English

In Table 3a we show the results from testing our unsupervised models on the English dataset. In Table 3b we show the results from testing our supervised models trained on the combined English and Norwegian dataset on the English STS data, while Table 3c shows the results for supervised models trained only on Norwegian data.

Since we have automatically translated the STS data, we are unsure how accurate the ground truth labels in Norwegian will be, since there will be examples of sentences where the similarity of the sentences changes because of differing translations. However we think that this should not influ-

| Model | STS12 | STS13 | STS14 | STS15 | STS16 | STSB | SICKR | Avg. |
|---|---|---|---|---|---|---|---|---|
| BERT(english) | 54.76 | 70.77 | 57.39 | 69.32 | 69.19 | 61.66 | 66.29 | 64.20 |
| roBERTa(english) | 65.26 | 77.06 | 67.09 | 76.88 | 76.71 | 75.32 | 65.60 | 71.99 |
| mBERT | 63.56 | 73.10 | 63.95 | 74.67 | 73.56 | 68.58 | 61.61 | 68.43 |
| WikiBERT | 64.68 | 77.60 | 67.04 | 76.20 | 76.30 | 74.63 | 65.34 | 71.68 |
| NorBERT | 52.96 | 62.30 | 54.99 | 67.45 | 69.83 | 63.68 | 62.40 | 61.94 |
| NB-BERT-base | 56.23 | 72.06 | 57.93 | 68.71 | 71.09 | 67.25 | 61.63 | 64.99 |
| NB-BERT-large | **72.54** | **83.68** | **76.08** | **83.03** | **81.09** | **81.32** | **68.80** | **78.08** |

(a) Performance of unsupervised models on English STS datasets.

| Model | STS12 | STS13 | STS14 | STS15 | STS16 | STSB | SICKR | Avg. |
|---|---|---|---|---|---|---|---|---|
| mBERT | 76.88 | 79.69 | 77.58 | 84.99 | 78.52 | 81.36 | 77.30 | 79.47 |
| WikiBERT | 72.45 | 59.56 | 67.08 | 80.87 | 75.21 | 75.31 | 74.01 | 72.07 |
| NorBERT | 73.39 | 69.40 | 72.65 | 83.10 | 77.30 | 80.48 | 76.55 | 76.13 |
| NBBert-base | 76.93 | 78.78 | 77.76 | 85.28 | 80.29 | 82.96 | 78.49 | 80.07 |
| NBBert-large | **78.30** | **85.92** | **81.78** | **87.11** | **83.24** | **85.72** | **79.56** | **83.09** |

(b) Performance of supervised models on English STS datasets fine-tuned on both Norwegian and English MNLI.

| Model | STS12 | STS13 | STS14 | STS15 | STS16 | STSB | SICKR | Avg. |
|---|---|---|---|---|---|---|---|---|
| mBERT | 72.62 | 79.36 | 75.84 | 81.87 | 79.70 | 77.48 | 70.18 | 76.72 |
| WikiBERT | 65.47 | 65.30 | 67.40 | 76.86 | 73.12 | 68.91 | 60.59 | 68.24 |
| NorBERT | 66.90 | 68.62 | 69.63 | 79.35 | 76.23 | 73.38 | 69.66 | 71.97 |
| NBBert-base | 71.57 | 80.30 | 76.30 | 81.55 | 79.23 | 78.09 | 71.12 | 76.88 |
| NBBert-large | **76.42** | **85.58** | **81.23** | **85.49** | **83.21** | **83.15** | **75.04** | **81.45** |

(c) Performance of supervised models on English STS datasets fine-tuned on Norwegian MNLI.

Table 3: Results of our models tested on the English STS datasets(Spearman's correlation).

ence comparisons between different models very much. This is supported by the fact that the internal ranking between models for the Norwegian and the English dataset is the same among the Norwegian unsupervised models. (English models unsurprisingly are higher in the rankings when tested on English)

One of the more interesting findings in this paper is how strong performance our models get on the English STS data. NB-BERT-base was initialized from the mBERT checkpoint which can partly explain this, but not all models was started from a model pre-trained in English. The unsupervised NB-BERT-large achieves a score of 78.08 on English STS. For comparison, the best unsupervised model in the original SimCSE paper, SimCSE-RoBERTa-large, achieved a score of 78.90. Thus we see that we have a model pre-trained on a Norwegian corpus (containg some English), further trained unsupervised in Norwegian, that achieves less than 1% worse score than the best English model, trained in English. This model is also better than the best unsupervised English model in the original SentenceBERT pa-

per. The supervised NB-BERT trained only on Norwegian NLI achieved a score of 81.45, while the version trained on Norwegian and English NLI achieve a score of 83.09. Comparably the supervised original English version SimCSE-BERT-base got a score of 81.57 and SimCSE-RoBERTa-large 83.76. Thus we see that we achieve comparable performance between a supervised Norwegian large BERT and a supervised English base BERT, when testing in English. Our best supervised model is less than 1% away from the best English SimCSE model, although this is less surprising than for the unsupervised models, since we in this case fine-tune our model also on English NLI. We also note that our best supervised model which is trained on only Norwegian is better than the best supervised English model in the Sentence-BERT paper. Thus it does seem like the models learn a lot for performing well at English sentence similarity even though the pre-training is mostly in Norwegian. The strong performance in English of NB-BERT models was already noted in (Kummervold et al., 2021).

To see if we can better understand the

| | |
|---|---|
| BERT | 76.7 |
| RoBERTa | 79.8 |
| mBERT | 80.2 |
| WikiBERT | 83.2 |
| NorBERT | 83.9 |
| NB-BERT-base | 82.7 |
| NB-BERT-large | 89.7 |

(a) Performance of unsupervised models when fine-tuned on the Talk of Norway dataset.

| | |
|---|---|
| mBERT | 79.3 |
| WikiBERT | 82.6 |
| NorBERT | 85.7 |
| NB-BERT-base | 83.4 |
| NB-BERT-large | 89.3 |

(b) Performance of supervised models trained on Norwegian NLI when fine-tuned on the Talk of Norway dataset.

| | |
|---|---|
| mBERT | 79.2 |
| WikiBERT | 81.1 |
| NorBERT | 84.9 |
| NB-BERT-base | 83.3 |
| NB-BERT-large | 89.3 |

(c) Performance of supervised models trained in on Norwegian and English NLI on the Talk of Norway dataset.

Table 4: Performance of our models on the ToN dataset(F1 score).

| | |
|---|---|
| BERT | 63.1 |
| RoBERTa | 64.4 |
| mBERT | 70.3 |
| WikiBERT | 77.0 |
| NorBERT | 82.0 |
| NB-BERT-base | 84.3 |
| NB-BERT-large | 87.6 |

(a) Performance of unsupervised models, fine-tuned on the NoReC binary classification dataset.

| | |
|---|---|
| mBERT | 72.2 |
| WikiBERT | 77.9 |
| NorBERT | 82.4 |
| NB-BERT-base | 85.9 |
| NB-BERT-large | 87.0 |

(b) Performance of supervised models trained on only Norwegian NLI when fine-tuned on the NoReC binary classification dataet.

| | |
|---|---|
| mBERT | 74.4 |
| WikiBERT | 77.6 |
| NorBERT | 81.0 |
| NB-BERT-base | 84.9 |
| NB-BERT-large | 87.3 |

(c) Performance of supervised models trained on Norwegian and English NLI when fine-tuned on the NoReC binary classification dataset.

Table 5: Performance of our models on the NoReC binary classification dataset(F1 score).

above findings, we tested the English supervised SimCSE-RoBERTa-large on Norwegian STS, and achieved only an average score of 54.23. Thus a very good English model scores badly in Norwegian, while a very good Norwegian model scores well in English. This might indicate that the reason the Norwegian models all perform so well in English is that there is enough English in the Norwegian training data (probably including many snippets in the Norwegian parts) that the models learn quite a lot of English.

# 6 Results classification

We report macro F1 score for the binary classification tasks.

## 6.1 ToN binary classification

In Table 4a we see the performance of the unsupervised models when fine-tuned on the Talk of Norway dataset. In Table 4b we see the performance of the supervised models trained on Norwegian NLI and then fine-tuned on the ToN dataset, while Table 4c shows the performance when training on both Norwegian and English NLI.

We see that training the models to give better sentence embeddings gives some performance gains on this task, compared to fine-tuning the base model: In (Kummervold et al., 2021) it is reported that NB-BERT achieves a score of 81.8, while NorBERT scores 78.2 and mBERT 78.4 on this task. All our numbers are slightly higher.

We see that for this classification task training to make sentence models with English NLI data included did not help: the numbers are very similar with and without it.

## 6.2 NoReC binary classification

In Table 5a we see the performance of unsupervised models on the NoReC binary classification task. In Table 5b we see the results of supervised models trained on Norwegian NLI, while in Table 5c we see the results of supervised models trained on Norwegian and English NLI.

For this task it is less clear that we get gains from training sentence embedding models: The highest reported number for this task is NB-BERT-base which is reported as 86.4 in (Kummervold

et al., 2021) and 83.9 in (Kutuzov et al., 2021). Our best score for NB-BERT-base is 85.9, which is not better than this. Our best model NB-BERT-large also does not achieve a higher score than about 87%, which is only slightly better than the smaller models. We do not know the reason we get improvements for ToN classification, and not here. The mBERT model do improve with training, but that is not so surprising, since it is not already as strong in Norwegian as most of the other models.

## 7 Discussion

We believe that our models perform well on the semantic sentence similarity task, even if we do not have any strict comparison since this is the first evaluation of Norwegian sentence embedding models on the STS data. The Norwegian dataset corresponds to the English one, so the scores of English models on English STS and Norwegian models on Norwegian STS should in principle correspond to each other, but because of the extra noise added by the automatic translation we are not surprised that the Norwegian numbers are a bit worse. We see that the models improve a lot compared to before training, and because they perform quite well even for the English STS datasets, we are confident that they have indeed learned something useful in Norwegian.

The supervised models perform better than our unsupervised models even though the supervised models are trained on machine translated data. This shows that machine translated data could be useful for doing NLP in smaller languages, at least for some tasks such as ours. The difference in the numbers we get for unsupervised and supervised training are similar to the ones in the original SimCSE paper. It is a bit unclear to what extent the specific content and language of the training data is important for performing well on STS tasks. For example, one can improve the performance of English SimCSE by training on unrelated image data (Jian et al., 2022). This might be because the task is a form of clustering, and images and text in other languages are structurally similar enough that the models learn something useful.

From doing our experiments we get comparisons of the different Norwegian language models. This is because this method of making sentence embeddings is mostly a way of extracting the knowledge already learned by the models, since the amount of training we do is much smaller than the amount the models already have been pre-trained. An unsuprising conclusion is that the scale of the model is the most important factor in making good language models. NB-BERT-large is the best model by clear margins for all of our evaluations. This conforms to the general tendency in recent NLP that scaling up models is more effective than tailoring data or architecture on a given scale. Next, we find that for binary classification the models NB-BERT-base and NorBERT perform quite similar, while WikiBERT is generally a bit weaker, while all of them clearly outperform mBERT. For sentence similarity we find different rankings among models: Here unsupervised WikiBERT is the second best model, while the supervised version is the weakest of the Norwegian supervised models. Supervised NB-BERT-base is clearly the second best model, while NorBERT performs worse on the STS task.

We see that training sentence embedding models slightly improves performance on the binary classification tasks, but not by much compared with the base models. There is no clear tendency on whether training supervised or unsupervised improves performance on classification more, since the numbers we get are similar in both cases.

## Acknowledgements

We are very grateful to Per Egil Kummervold of the National Library of Norway's AI lab for helpful conversations, as well as for sharing the translated MNLI dataset.

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
