# OpenReview forum: "Training and Evaluating Norwegian Sentence Embedding Models"
_NoDaLiDa/2023/Conference — NoDaLiDa 2023_

### Official Review · Reviewer_VLrp · 2023-02-28
**An obvious application but a good paper setting benchmarks for Norwegian Sentence Embeddings.**

**Rating:** 7
**Confidence:** 4

**Review:**

This paper is very much a replica of the work presented by Gao et al (2021) but applied to Norwegian rather than English. The same method, SimCSE, developed in that paper, is used and, to a large extent, the same test sets. A twist is the use of MT for producing test sets in Norwegian and some of the training data for the supervised training. Another difference is that the paper also tests the generated sentence embeddings on two classification tasks.

The quality of the work is high and so is the clarity of the paper. The originality is low to fair and the significance hard to evaluate. On the positive side the paper bases its analyses on, as far as I can tell, all available BERT-based models for Norwegian, which makes comparisons possible, and by evaluating also on English data, also makes comparisons with Gao et al's results possible. An interesting result is that their results for English test data are compatible with Gao et al's, even though some systems were largely trained on Norwegian. Thus they support the claim that there models, and especially the best model, are good models. One can also conclude from this that the methods would work well for other Scandinavian languages.

The test results are largely consistent across test data and systems, but not completely. The largest shortcoming of the paper is the lack of explanations for the results, apart from the expected that 'large' performs better than 'base', and supervised training improves on the unsupervised models.

**Paper Type:**

Long paper

---

### Official Review · Reviewer_iTm4 · 2023-03-07
**The research presents and evaluates several sentence transformer models trained for the Norwegian (sometimes together with English) language.**

**Rating:** 8
**Confidence:** 4

**Review:**

The research is interesting and clearly described. Moreover, the novelty of the research is emphasized. My remarks:
1.	The Related Work section focuses only on what is done for the Norwegian language and on the authors’ research. It would be interesting to see the broader picture here (i.e., to have a summary of what is done for other languages on this topic).
2.	In your research you train primary BERT-based models (lines 298-310) under SimCSE settings to get sentence transformers. However, there are already pre-trained sentence transformer models (including multilingual and even cross-lingual ones that also support Norwegian) and some of them are very sophisticated (e.g., LaBSE). It would be very interesting to see how your results compare to LaBSE (and other sentence transformer models) in sentence similarity and classification problems. This task could be for your future research.
3.	Do you have any explanation why the NB-BERT-large model (which is trained from scratch on the Norwegian texts as said in https://huggingface.co/NbAiLab/nb-bert-large) performs so well also for the English language? Maybe the reason is that both languages are somehow similar (shared vocabulary, strict sentence structure?). My remark is from the analysis of Table 2 and Table 3.
4.	Are you using the macro F1 score in Table 2 and Table 3?
5.	Please, translate the examples from Figure 2 (they do not make any sense to people not understanding Norwegian).

**Paper Type:**

Long paper

---

### Decision · Program_Chairs · 2023-03-17

Accept